# PROVABLY ROBUST COST-SENSITIVE LEARNING VIA RANDOMIZED SMOOTHING

## ABSTRACT

We focus on learning adversarially robust classifiers under a cost-sensitive scenario, where the potential harm of different classwise adversarial transformations is encoded in a binary cost matrix. Existing methods are either empirical that cannot certify robustness or suffer from inherent scalability issues. In this work, we study whether randomized smoothing, a scalable certification framework, can be leveraged to certify cost-sensitive robustness. Built upon a notion of cost-sensitive certified radius, we show how to adapt the standard randomized smoothing certification pipeline to produce tight robustness guarantees for any given cost matrix. In addition, with fine-grained certified radius optimization schemes designed for different data subgroups, we propose an algorithm to train smoothed classifiers that are optimized for cost-sensitive robustness. Extensive experiments on image benchmarks and a real-world medical dataset demonstrate the superiority of our method in achieving significantly improved performance of certified cost-sensitive robustness while having a negligible impact on overall accuracy.

## 1 INTRODUCTION

Recent studies have revealed that deep learning models are highly vulnerable to adversarial examples (Szegedy et al., 2013; Goodfellow et al., 2014). To improve model robustness in the presence of adversarial examples, various defensive mechanisms have been proposed, primarily falling into two categories: *empirical defenses* (Goodfellow et al., 2014; Papernot et al., 2016; Kurakin et al., 2016; Madry et al., 2017; Zhang et al., 2019) and *certifiable methods* (Raghunathan et al., 2018; Wong & Kolter, 2018; Gowal et al., 2018; Cohen et al., 2019; Lecuyer et al., 2019; Jia et al., 2019; Li et al., 2019). In particular, certifiable methods can produce a robustness certificate for the model prediction to remain unchanged within some specific norm-bounded perturbation ball of any testing input and train models to be provably robust with respect to the certificate.

Most existing defenses aim to improve the overall robustness of a classification model, assuming the same penalty is imposed on all kinds of adversarial misclassifications. For real-world applications, however, it is likely that some specific misclassifications are more consequential than others (Domingos, 1999; Elkan, 2001). For instance, misclassifying a malignant tumor as benign in the application of medical diagnosis is much more detrimental to a patient than the reverse. Therefore, instead of solely focusing on enhancing overall robustness, the development of defenses should also account for the difference in costs induced by different adversarial examples. In line with existing works on cost-sensitive robust learning (Domingos, 1999; Asif et al., 2015; Zhang & Evans, 2019; Chen et al., 2021), we aim to develop models that are robust to cost-sensitive adversarial misclassifications, while maintaining the standard overall classification accuracy. However, existing defenses are either hindered by their foundational reliance on heuristics, which often fall short of providing a robustness guarantee (Domingos, 1999; Asif et al., 2015; Chen et al., 2021), or suffer from inherent scalability issues (Zhang & Evans, 2019) (see detailed discussions of related works in Appendix A).

To achieve the best of both worlds, we propose to learn provably cost-sensitive robust classifiers by leveraging randomized smoothing (Liu et al., 2018; Cohen et al., 2019; Salman et al., 2019), an emerging robustness certification framework that has attracted a lot of attention due to its simplicity and scalability. However, optimizing smoothed classifiers for cost-sensitive robustness is intractable, primarily due to the inherent complexity introduced by the discrete Gaussian sampling process when

transforming base classifiers into smoothed classifiers. In addition, different designs of cost matrix necessitate a more flexible and targeted optimization scheme than optimizing for overall robustness.

**Contributions.** We are the first to adapt the randomized smoothing framework to certify and train for cost-sensitive robustness. In particular, for any binary cost matrix (Section 2), we introduce the notion of *cost-sensitive certified radius* (Definition 3.1), which captures the maximum allowable $\ell_2$ perturbation with respect to the smoothed classifier for each sensitive seed input (Theorem 3.2). We show that compared with standard certified radius typically adopted in the literature, our proposed definition is more suitable for certifying cost-sensitive robustness, as it theoretically defines a larger certified radius specifically for a wide range of cost matrices (Theorem 3.3). Built upon the definition of cost-sensitive certified radius, we further propose a practical certification algorithm using Monte Carlo samples (Algorithm 1), which provides two different methods for computing the probabilistic bound on certified radius such that the tighter bound can always be returned (Section 3.2). Moreover, to train for cost-sensitive robust smoothed classifiers, we attempt to adapt the commonly-used reweighting method in standard cost-sensitive learning literature (Elkan, 2001; Khan et al., 2017). However, we discover that the naive reweighting scheme does not adapt well when training a smoothed classifier, owing to the indirect optimization of the base classifier and the non-optimal trade-off between sensitive and non-sensitive examples (Section 4.1). Therefore, we take the distinctive properties of different data subgroups into account and design an advanced cost-sensitive robust training method based on MACER (Zhai et al., 2020) to directly optimize the certified radius with respect to the smoothed classifier (Section 4.2). Experiments on typical image benchmarks and a real-world medical dataset illustrate the superiority of our method in achieving consistently and significantly improved certified cost-sensitive robustness while maintaining a similar performance of overall accuracy under a variety of cost matrix settings (Section 5).

## 2 PRELIMINARIES

**Randomized Smoothing.** Cohen et al. (2019) proposed a probabilistic certification framework, named as randomized smoothing (RS), which is able to certify robustness for large-scale models. In particular, randomized smoothing is based on the following construction of smoothed classifiers, which first augments normal inputs with randomly sampled Gaussian noise, then passes the noisy inputs through a base classifier and aggregate their predictions using majority voting:

**Definition 2.1.** Let $\mathcal{X} \subseteq \mathbb{R}^d$ be the input space and $[m] := \{1, 2, \ldots, m\}$ be the label space. For any base classifier $f_\theta : \mathcal{X} \to [m]$ and $\sigma > 0$, the corresponding *smoothed classifier* $g_\theta$ is defined as:

$$g_\theta(\boldsymbol{x}) = \underset{k \in [m]}{\arg\max} \, \mathbb{P}_{\boldsymbol{\delta} \sim \mathcal{N}(\boldsymbol{0}, \sigma^2 \mathbf{I})} \big[ f_\theta(\boldsymbol{x} + \boldsymbol{\delta}) = k \big], \, \forall \boldsymbol{x} \in \mathcal{X}.$$

Let $h_\theta : \mathcal{X} \to [0, 1]^m$ be the function that maps any input $\boldsymbol{x}$ to the prediction probabilities of $g_\theta(\boldsymbol{x})$:

$$[h_\theta(\boldsymbol{x})]_k = \mathbb{P}_{\boldsymbol{\delta} \sim \mathcal{N}(\boldsymbol{0}, \sigma^2 \mathbf{I})} \big[ f_\theta(\boldsymbol{x} + \boldsymbol{\delta}) = k \big], \, \forall \boldsymbol{x} \in \mathcal{X} \text{ and } k \in [m].$$

The following lemma, proven in Cohen et al. (2019), characterizes the maximum allowable $\ell_2$-perturbation for any input $\boldsymbol{x}$ such that the prediction of $g_\theta$ at $\boldsymbol{x}$ remains the same within the radius.

**Lemma 2.2** (Cohen et al. (2019)). *Let $\boldsymbol{x} \in \mathcal{X}$ be any input and $y \in [m]$ be its ground-truth label. If $g_\theta$ classifies $\boldsymbol{x}$ correctly, i.e., $y = \arg\max_{k \in [m]} \mathbb{P}_{\boldsymbol{\delta} \sim \mathcal{N}(\boldsymbol{0}, \sigma^2 \mathbf{I})} \big[ f_\theta(\boldsymbol{x} + \boldsymbol{\delta}) = k \big]$, then the prediction of $g$ at $\boldsymbol{x}$ is both accurate and provably robust with certified $\ell_2$-norm radius $R(\boldsymbol{x})$, which is defined as:*

$$R(\boldsymbol{x}) = \frac{\sigma}{2} \Big[ \Phi^{-1} \big( [h_\theta(\boldsymbol{x})]_y \big) - \Phi^{-1} \big( \max_{k \neq y} [h_\theta(\boldsymbol{x})]_k \big) \Big], \tag{1}$$

*where $\Phi$ is the CDF of standard Gaussian $\mathcal{N}(0, 1)$ and $\Phi^{-1}$ denotes its inverse.*

We remark that our method is built upon this randomized smoothing framework, which is designed for $l_2$ perturbations. Recent studies have adapted the standard framework to certify robustness against other $\ell_p$-norm bounded perturbations (Mohapatra et al., 2020; Yang et al., 2020). We believe our method is applicable to those $\ell_p$-norms, whereas studying how to certify cost-sensitive robustness against general perturbations defined in metrics beyond $\ell_p$-norm would be an interesting future work.

**Cost-Sensitive Robustness.** We consider robust classification tasks under cost-sensitive scenarios, where the goal is to learn a classifier with both high overall accuracy and cost-sensitive robustness. Suppose $\mathbf{C} \in \{0,1\}^{m \times m}$ is a predefined cost matrix that encodes the potential harm of different classwise adversarial transformations.[1] In particular, the most studied overall robustness corresponds to having a cost matrix, where all the entries are 1 except for the diagonal ones. For any $j \in [m]$ and $k \in [m]$, $C_{jk} = 1$ means any misclassification from seed class $j$ to target class $k$ will induce a cost, whereas $C_{jk} = 0$ suggests that there is no incentive for an attacker to trick the model to misclassify inputs from class $j$ to class $k$. Therefore, the goal of cost-sensitive robust learning is to reduce the number of adversarial misclassifications that will induce a cost defined by $\mathbf{C}$. For the ease of presentation, we also introduce the following notations. For any seed class $j \in [m]$, we let $\Omega_j = \{k \in [m] : C_{jk} = 1\}$ be the set of cost-sensitive target classes. If $\Omega_j$ is an empty set, then all the examples from seed class $j$ is non-sensitive. Otherwise, any class $j$ with $|\Omega_j| \geq 1$ is a sensitive seed class. Given a dataset $\mathcal{S} = \{(\boldsymbol{x}_i, y_i)\}_{i \in [n]}$, we define the set of cost-sensitive examples as $\mathcal{S}^{\mathrm{s}} = \{(\boldsymbol{x}, y) \in \mathcal{S} : |\Omega_y| \geq 1\}$, while the remaining examples are regarded as non-sensitive.

## 3 CERTIFYING COST-SENSITIVE ROBUSTNESS

This section explains how to certify cost-sensitive robustness using randomized smoothing. We first introduce the formal definition of cost-sensitive certified radius and the corresponding evaluation metrics then discuss its connection to the standard certified radius (Section 3.1). Finally, based on the proposed definition, we design a practical certification algorithm using finite samples (Section 3.2).

### 3.1 COST-SENSITIVE CERTIFIED RADIUS

Recall that for any example $(\boldsymbol{x}, y) \in \mathcal{S}^{\mathrm{s}}$, only misclassifying $\boldsymbol{x}$ to a target class in $\Omega_y$ incurs a cost, whereas misclassifications to any class from $[m] \backslash \Omega_y$ is tolerable. Below, we formally define *cost-sensitive certified radius*, which adapts the standard certified radius to cost-sensitive scenarios:

**Definition 3.1** (Cost-Sensitive Certified Radius). Consider the same setting as in Theorem 2.2. Let $\mathbf{C}$ be an $m \times m$ cost matrix. For any example $(\boldsymbol{x}, y) \in \mathcal{X} \times [m]$ where $y$ is a sensitive seed class, the *cost-sensitive certified radius* at $(\boldsymbol{x}, y)$ with respect to $\mathbf{C}$ is defined as:

$$R_{\text{c-s}}(\boldsymbol{x}; \Omega_y) = \frac{\sigma}{2} \left[ \Phi^{-1}\Big( \max_{k \in [m]} \big[h_\theta(\boldsymbol{x})\big]_k \Big) - \Phi^{-1}\Big( \max_{k \in \Omega_y} \big[h_\theta(\boldsymbol{x})\big]_k \Big) \right],$$

where $\Omega_y = \{k \in [m] : C_{yk} = 1\}$, and $\Phi^{-1}$ is the inverse CDF of standard Gaussian $\mathcal{N}(0, 1)$.

Based on Definition 3.1, the following theorem extends Lemma 2.2, which shows how to produce a certificate for cost-sensitive robustness of a smoothed classifier with respect to any given input.

**Theorem 3.2.** *Consider the same setting as in Definition 3.1. For any example $(\boldsymbol{x}, y)$, if the predicted class of the smoothed classifier $g_\theta$ at $\boldsymbol{x}$ does not incur a cost, i.e., $\arg\max_{k \in [m]}[h_\theta(\boldsymbol{x})]_k \notin \Omega_y$, then $g_\theta$ is provably robust at $\boldsymbol{x}$ with certified radius $R_{\text{c-s}}(\boldsymbol{x}; \Omega_y)$ measured in $\ell_2$-norm.*

Theorem 3.2 can be applied to certify cost-sensitive robustness for any binary-valued cost matrix. Note that $\max_{k \in [m]}[h_\theta(\boldsymbol{x})]_k$, the first term on the right hand side of Definition 3.1, denotes the maximum predicted probability across all classes with respect to $h_\theta$, while the second term $\max_{k \in \Omega_y}[h_\theta(\boldsymbol{x})]_k$ is the maximum predicted probability across all sensitive target classes within $\Omega_y$, which are different from the corresponding terms used in standard certified radius (Equation 1). We remark that these modifications are necessary for providing tight certification results for cost-sensitive settings, since it is possible for the prediction $g(\boldsymbol{x})$ to be incorrect but cost-sensitive robust under certain scenarios, as long as the incorrect prediction does not fall into the set of sensitive target classes $\Omega_y$.

More specifically, the following theorem, proven in Appendix B, characterizes the connection between the cost-sensitive certified radius and the standard notion of certified radius for any cost matrix.

**Theorem 3.3.** *For any cost-sensitive scenario, if the prediction $g_\theta(\boldsymbol{x})$ does not incur a cost, then $R_{\text{c-s}}(\boldsymbol{x}; \Omega_y) \geq R(\boldsymbol{x})$, where the equality holds when $\Omega_y = \{k \in [m] : k \neq y\}$.*

---

[1]Although we only consider binary-valued cost matrices in this work, our method can be easily adapted to provide cost-sensitive robustness guarantees for real-valued cost matrices (Zhang & Evans, 2019).

---

**Algorithm 1** Certification for Cost-Sensitive Robustness

1: **function** CERTIFY($f, \sigma, \boldsymbol{x}, n_0, n, \alpha, \Omega_y$)
2: $\quad$ count$_0 \leftarrow$ SAMPLEUNDERNOISE($f, \boldsymbol{x}, n_0, \sigma$)
3: $\quad \hat{c}_A \leftarrow$ top index in count$_0$
4: $\quad$ count $\leftarrow$ SAMPLEUNDERNOISE($f, \boldsymbol{x}, n, \sigma$)
5: $\quad \underline{p_A} \leftarrow$ LowerConfBnd(count[$\hat{c}_A$], $n, 1 - \alpha$)
6: $\quad R_1 = \sigma \Phi^{-1}(\underline{p_A})$
7: $\quad \hat{c}_B \leftarrow$ top index in count[$\Omega_y$]
8: $\quad \underline{p_A} \leftarrow$ LowerConfBnd(count[$\hat{c}_A$], $n, 1 - \alpha/2$)
9: $\quad \overline{p_B} \leftarrow$ UpperConfBnd(count[$\hat{c}_B$], $n, 1 - \alpha/(2|\Omega_y|)$)
10: $\quad R_2 = \frac{\sigma}{2}(\Phi^{-1}(\underline{p_A}) - \Phi^{-1}(\overline{p_B}))$
11: $\quad$ **if** $\max(R_1, R_2) > 0$
12: $\quad\quad$ **return** prediction $\hat{c}_A$ and $\max(R_1, R_2)$
13: $\quad$ **else**
14: $\quad\quad$ **return** ABSTAIN
15: **end function**

---

Theorem 3.3 suggests that using $R_{\text{c-s}}(\boldsymbol{x}; \Omega_y)$ can always yield a cost-sensitive robustness certificate not inferior to that using the standard certified radius $R(\boldsymbol{x})$. In particular, for input $\boldsymbol{x}$ with $|\Omega_y| \neq m - 1$, the improvement of cost-sensitive robustness certification based on $R_{\text{c-s}}(\boldsymbol{x}; \Omega_y)$ is likely to be more significant. As will be shown in Figures 1(a) and 1(b), the empirically estimated robustness certificate with $R_{\text{c-s}}(\boldsymbol{x}; \Omega_y)$ consistently surpasses that with $R(\boldsymbol{x})$ across various cost matrix settings.

**Evaluation Metrics.** We formally define our evaluations metrics, *certified cost-sensitive robustness* and *overall accuracy*, which will be used to measure a classifier's performance under a cost-sensitive setting. For any binary cost matrix, we define *certified cost-sensitive robustness* as the ratio of cost-sensitive examples that are provably robust with $g_\theta$ against $\ell_2$ perturbations with strength $\epsilon > 0$:

$$\text{Rob}_{\text{c-s}}(g_\theta) = \frac{1}{|\mathcal{S}^\text{s}|} \sum_{(\boldsymbol{x}, y) \in \mathcal{S}^\text{s}} \mathbb{1}\{R_{\text{c-s}}(\boldsymbol{x}; \Omega_y) > \epsilon\}, \text{ provided } |\mathcal{S}^\text{s}| > 0, \tag{2}$$

where $\mathcal{S}^\text{s}$ denotes the set of cost-sensitive examples. In addition, the *overall accuracy* of $g_\theta$ is defined as the fraction of correctly classified samples with respect to the whole training dataset:

$$\text{Acc}(g_\theta) = \frac{1}{|\mathcal{S}|} \sum_{(x, y) \in \mathcal{S}} \mathbb{1}\{R(\boldsymbol{x}) > 0\}. \tag{3}$$

We use the standard certified radius $R(\boldsymbol{x})$ in Equation 3, since the computation of overall accuracy does not rely on the cost matrix $\mathbf{C}$. As will be discussed next, we are going to replace $\text{Rob}_{\text{c-s}}(g_\theta)$ and $R(\boldsymbol{x})$ with their empirical counterparts for practical implementations of the two metrics.

## 3.2 PRACTICAL CERTIFICATION ALGORITHM

By definition, the construction of $h_\theta$ requires access to an infinite number of Gaussian samples to compute the cost-sensitive certified radius. However, it is computationally infeasible in practice to obtain the exact value of $R_{\text{c-s}}(\boldsymbol{x}; \Omega_y)$ even for a single example $(\boldsymbol{x}, y)$. The difference between $R_{\text{c-s}}(\boldsymbol{x}; \Omega_y)$ and standard certified radius $R(\boldsymbol{x})$ necessitates a new certification procedure. In this section, we put forward a new Monte Carlo algorithm for certifying cost-sensitive robustness by adapting Cohen et al. (2019)'s standard algorithm, which is applicable to any binary cost matrix.

Algorithm 1 depicts the pseudocode of our certification method, which involves two ways to compute the confidence bound for cost-sensitive certified radius $R_{\text{c-s}}(\boldsymbol{x}; \Omega_y)$ (see Appendix C for more details about the sampling scheme). The first approach is based on $R_1$, which is computed using a lower $1 - \alpha$ confidence bound on the ground-truth probability $p_A$ with respect to the top class, same as the bound for standard radius used in Cohen et al. (2019) for certifying overall robustness. Note that according to Definition 3.1, the standard certified radius is guaranteed to be less than or equal to the corresponding cost-sensitive certified radius, suggesting that $R_1$ also severs as a valid $1 - \alpha$ confidence bound for cost-sensitive radius. On the other hand, $R_2$ is computed using both a lower

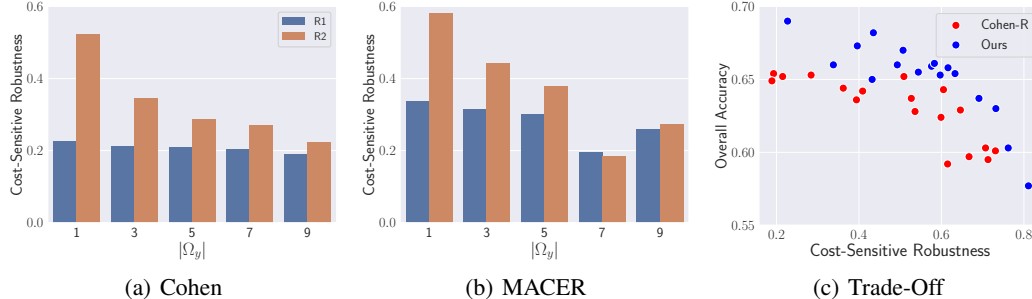

(a) Cohen  (b) MACER  (c) Trade-Off

Figure 1: Comparisons of certified cost-sensitive robustness computed using $R_1$ and $R_2$ with respect to (a) Cohen and (b) MACER. Figure 1(c) visualizes the trade-off between overall accuracy and cost-sensitive robustness with respect to our method and Cohen-R. We fix the CIFAR-10 class "cat" as the only sensitive seed class and vary the corresponding cost-sensitive target classes.

$1 - \alpha/2$ confidence bound of $p_A$ and an upper $1 - \alpha/2|\Omega_y|$ bound of $p_B$, the ground-truth probability of the top class within $\Omega_y$. The following theorem, proven in Appendix C by union bound, shows that the confidence for $R_2$ being a cost-sensitive certified radius is guaranteed to be at least $1 - \alpha$.

**Theorem 3.4.** *For any example $(\boldsymbol{x}, y)$, the second radius $R_2$ specified by Algorithm 1 is a certified cost-sensitive robust radius with at least $1 - \alpha$ confidence over the randomness of Gaussian sampling.*

Theorem 3.4 suggests that the prediction of $g_\theta$ at $(\boldsymbol{x}, y)$ will not incur any undesirable cost with high probability as long as the $\ell_2$ perturbation is within radius $\max(R_1, R_2)$. By definition, the output $\max(R_1, R_2)$ of Algorithm 1 is always better than $R_1$ solely for certifying cost-sensitive robustness.

**Comparison between $R_1$ and $R_2$.** Note that $R_2$ is specifically designed for certifying cost-sensitive robustness, whereas $R_1$ works for both overall and cost-sensitive scenarios. To further illustrate the superiority of our proposed practical certification algorithm to Cohen et al. (2019)'s, we specify the scenarios where $R_2$ is superior to $R_1$ by definition and provide empirical evidence in Figures 1(a) and 1(b). By definition, there are two scenarios where $R_2$ is likely to be larger than $R_1$:

1. $\Omega_y$ does not contain the second highest probability class, which is likely to happen when the number of cost-sensitive target classes is small, i.e., $|\Omega_y| = 1$.

2. Even when $|\Omega_y| = m - 1$, it is still possible for $R_2 > R_1$, especially if the ground-truth probability of the second highest class $p_B$ is far below $1 - p_A$.

In summary, if $p_B$, the ground-truth probability of the top class in $\Omega_y$, is far below $p_A$, then $R_2$ will be much higher than $R_1$, thus producing a much tighter cost-sensitive robust certificate. To further study the relationship between $R_1$ and $R_2$, we conduct experiments to compare the cost-sensitive robustness computed based on $R_1$ and $R_2$ across different cost matrix scenarios, as shown in Figures 1(a) and 1(b). We fix a single seed class "cat" as sensitive and vary the size of its corresponding cost-sensitive target classes, randomly selecting $|\Omega_y| \in \{1, 3, 5, 7, 9\}$ of all the possible target classes. We then evaluate the performance of the models produced by two baseline training methods for randomized smoothing, Cohen (Cohen et al., 2019) and MACER (Zhai et al., 2020). Figures 1(a) and 1(b) show that cost-sensitive robustness measured by $R_2$ outperforms that of $R_1$ across most of the settings. As the size of $\Omega_y$ decreases, the performance gap between $R_1$ and $R_2$ is more pronounced, underscoring the superior efficacy of $R_2$ and confirming the superiority of Algorithm 1 over Cohen et al. (2019)'s.

## 4 TRAINING FOR COST-SENSITIVE ROBUSTNESS

A popular training scheme for cost-sensitive learning is reweighting (Elkan, 2001), which assigns larger weights to cost-sensitive inputs during model training. Thus, a natural question is whether the reweighting scheme can be incorporated in randomized smoothing to train for cost-sensitive robustness. In this section, we first study the effectiveness of reweighting methods combined with the method of Cohen et al. (2019) (Section 4.1), then provide a new training method by leveraging the design insight of MACER (Zhai et al., 2020) to achieve better performance (Section 4.2).

## 4.1 REWEIGHTING METHOD

We consider the base classifier training method introduced in Cohen et al. (2019), which proposes to inject Gaussian noise to all inputs during the training process of $f_\theta$. Given a binary cost matrix, $\mathcal{D}$ is the underlying data distribution, let $\mathcal{D}_s$ be the distribution of all sensitive examples which incur costs if misclassified and let $\mathcal{D}_n$ represent the distribution of the remaining normal examples. Intuitively, the training pipeline of randomized smoothing can be adapted to cost-sensitive settings using a simple reweighting scheme by increasing the weights assigned to the loss function of sensitive examples, denoted as Cohen-R. More concretely, the training objective of Cohen-R is defined as follows:

$$\min_{\theta \in \Theta} \left[ \mathbb{E}_{(\boldsymbol{x},y)\sim\mathcal{D}_n} \mathcal{L}_{CE}\big(f_\theta(\boldsymbol{x}+\boldsymbol{\delta}),y\big) + \alpha \cdot \mathbb{E}_{(\boldsymbol{x},y)\sim\mathcal{D}_s} \mathcal{L}_{CE}\big(f_\theta(\boldsymbol{x}+\boldsymbol{\delta}),y\big) \right],$$

where $\Theta$ denotes the set of model parameters, $\mathcal{L}_{CE}$ represents the cross-entropy loss, and $\alpha \geq 1$ is a trade-off parameter that controls the performance between sensitive and non-sensitive examples. When $\alpha = 1$, the above objective function is equivalent to the training loss used in standard randomized smoothing (Cohen et al., 2019). However, due to the indirect optimization of the smoothed classifier, Cohen-R tends to sacrifice overall accuracy to a large degree when trying to achieve high cost-sensitive robustness (see Figure 1(c) for empirical evidence about this statement).

## 4.2 OUR METHOD

In this section, we propose a more direct optimization scheme based on the proposed notion of certified cost-sensitive radius, leveraging a similar insight of MACER (Zhai et al., 2020) to better trade off cost-sensitive robustness and overall accuracy. To simplify notations, we first introduce a general class of margin-based losses. Given $l \leq u$ denoting the thresholding parameters, we define the following class of margin losses: for any $r \in \mathbb{R}$ representing the certified radius, let

$$\mathcal{L}_M\big(r;l,u\big) = \max\{u-r,0\} \cdot \mathbb{1}(l \leq r \leq u).$$

Here, the indicator function selects data points whose certified radius falls into the range of $[l,u]$. For any binary cost matrix $\mathbf{C}$, the total training loss function of our method is defined as:

$$\min_{\theta \in \Theta} I_1 + \lambda \cdot I_2 + \lambda \cdot I_3, \tag{4}$$

$$\text{where } I_1 = \mathbb{E}_{(\boldsymbol{x},y)\sim\mathcal{D}} \mathcal{L}_{CE}\big(h_\theta(\boldsymbol{x}),y\big),$$

$$I_2 = \mathbb{E}_{(\boldsymbol{x},y)\sim\mathcal{D}} \mathcal{L}_M\big(R_{\text{c-s}}(\boldsymbol{x};\Omega_y,h_\theta);0,\gamma_1\big),$$

$$I_3 = \mathbb{E}_{(\boldsymbol{x},y)\sim\mathcal{D}_s} \mathcal{L}_M\big(R_{\text{c-s}}(\boldsymbol{x};\Omega_y,h_\theta);-\gamma_2,\gamma_2\big),$$

where $\lambda, \gamma_1, \gamma_2 > 0$ are hyperparameters, $R_{\text{c-s}}(\boldsymbol{x};\Omega_y,h_\theta)$ is the cost-sensitive certified radius with respect to $h_\theta$, and $\mathcal{D}, \mathcal{D}_s$ denote the underlying distributions of all and cost-sensitive examples, respectively. In particular, Equation 4 consists of three terms: $I_1$ represents the cross-entropy loss with respect to $h_\theta$ over $\mathcal{D}$, which controls the overall accuracy; $I_2$ and $I_3$ control the robustness with a shared trade-off parameter $\lambda$.[2] The range of the interval $[l,r]$ represents which data subpopulation we want to optimize. A larger thresholding parameter such as $\gamma_1$ and $\gamma_2$ lead to a higher data coverage, whereas the range with a smaller threshold includes fewer data points. We set $\gamma_2 > \gamma_1$ to have a wider adjustment range for sensitive seed examples. As shown in Wang et al. (2020), optimizing misclassified samples can help adversarial robustness, thus, we intend to include sensitive seed examples with a negative radius in $[-\gamma_2, 0)$ in the design of $I_3$ for a better performance.

Intuitively speaking, by imposing different threshold restrictions $[l,u]$ on the certified radius of sensitive seed classes and normal seed classes, the optimization process can prioritize making adjustments to data subpopulations of specific classes rather than considering all data points belonging to those classes. This is also a key advantage of our method over the naive reweighting method. As will be shown in our experiments, such fine-grained optimization enables our method to improve certified cost-sensitive robustness to a large extent without sacrificing overall accuracy. Figure 1(c) provide empirical evidence of the non-optimal trade-off for the reweighting method *Cohen-R* on CIFAR-10 under a specific cost matrix of "cat" being the only sensitive seed class, compared with

---

[2]When $|\Omega_y| < m-1$, we optimize the data distribution $\mathcal{D}$ in $I_2$ to counteract $I_3$'s potential adverse effects. However, for $|\Omega_y| = m-1$, we utilize $\mathcal{D}_n$ since $I_2$ and $I_3$ have no optimization conflicts.

Table 1: Certification results for seedwise cost matrices. The noise level $\sigma$ is 0.5 for both CIFAR-10 and Imagenette. Acc stands for overall accuracy. $\text{Rob}_{\text{c-s}}$ is certified cost-sensitive robustness estimated by $\max(R_1, R_2)$, $\text{Rob}_{\text{std}}$ is certified cost-sensitive robustness estimated by $R_1$, $\text{Rob}_{\text{non}}$ is certified robustness of non-sensitive samples, all measured at $\epsilon = 0.5$. The best statistics are highlighted in bold.

| Dataset | Type | Method | Acc | $\text{Rob}_{\text{c-s}}$ | $\text{Rob}_{\text{std}}$ | $\text{Rob}_{\text{non}}$ |
|---|---|---|---|---|---|---|
| CIFAR-10 | Single (3) | Cohen | 0.654 | 0.223 | 0.193 | 0.466 |
| | | MACER | 0.659 | 0.273 | 0.259 | 0.499 |
| | | Cohen-R | 0.642 | 0.506 | 0.466 | 0.437 |
| | | Ours | **0.661** | **0.628** | 0.598 | 0.455 |
| | Multi (2, 4) | Cohen | 0.659 | 0.293 | 0.257 | 0.463 |
| | | MACER | 0.659 | 0.291 | 0.262 | 0.527 |
| | | Cohen-R | 0.662 | 0.348 | 0.308 | 0.413 |
| | | Ours | **0.669** | **0.461** | 0.441 | 0.504 |
| Imagenette | Single (7) | Cohen | **0.803** | 0.646 | 0.637 | 0.744 |
| | | MACER | 0.782 | 0.638 | 0.638 | 0.739 |
| | | Cohen-R | 0.746 | 0.733 | 0.733 | 0.671 |
| | | Ours | 0.791 | **0.811** | 0.795 | 0.725 |
| | Multi (3, 7) | Cohen | **0.804** | 0.589 | 0.579 | 0.771 |
| | | MACER | 0.782 | 0.578 | 0.578 | 0.774 |
| | | Cohen-R | 0.740 | 0.617 | 0.613 | 0.843 |
| | | Ours | 0.791 | **0.715** | 0.713 | 0.738 |

the performance of models produced by our proposed method. Each $(\text{Rob}_{\text{c-s}}, \text{Acc})$ point is derived from a unique set of model tuning hyperparameters. Notably, the points corresponding to our method predominantly occupy the upper-right quadrant compared to those of Cohen-R illustrating that our proposed approach offers a superior trade-off between cost-sensitive robustness and overall accuracy.

## 5 Experiments

We evaluate the performance of our method on two image benchmarks: CIFAR-10 (Krizhevsky et al., 2009) and Imagenette, a 10-class subset of ImageNet.[3] In addition, we test our method on the medical dataset HAM10k (Ghosh et al., 2023) to further examine its generalizability for real-world scenarios, where cost-sensitive misclassifications have more severe consequences. For CIFAR-10 and HAM10k, we use the same ResNet (He et al., 2016) architecture as employed in (Cohen et al., 2019). Specifically, we choose ResNet-56 network, since it attains comparable performance to ResNet-110 with a shorter computation time. For Imagenette, we use ResNet18 following (Pethick et al., 2023).

**Baselines.** We primarily compare our method with two prevalent randomized smoothing methods: Cohen (Cohen et al., 2019) and MACER (Zhai et al., 2020). We also consider Cohen-R, a variant of Cohen adapted for cost-sensitive scenarios using reweighting technique described in Section 4.1. We select Cohen for comparisons with standard randomized smoothing and MACER for comparing with methods that optimize for certified radius. Both of these baselines are optimized for overall robustness. Cohen-R is optimized for cost-sensitive performance by tuning the weight parameter $\alpha$ for a fair comparison. In addition, our experiments are mainly conducted for two categories of cost matrices: *seedwise* and *pairwise*, as their corresponding training procedures are slightly different.

**Experimental Details.** Tuning the hyperparameters plays a critical role. For Cohen-R, the weight parameters $\alpha$ are carefully tuned to ensure the best possible trade-off between overall accuracy and cost-sensitive robustness, where we enumerate all values from $\{1.0, 1.1, \ldots, 2.0\}$ and find that nearly in all cases of cost matrices, $\alpha = 1.2$ achieves the best result. For our method, we follow Zhai et al. (2020)'s settings for training. The main difference between our method and MACER is the choice of

---

[3]This dataset can be downloaded from `https://github.com/fastai/imagenette`.

Table 2: Certification results for pairwise cost matrices. $\epsilon = 0.5$ and $\sigma = 0.5$ for both CIFAR-10 and Imagenette. Acc stands for overall accuracy and $\mathrm{Rob_{c\text{-}s}}$ refers to certified cost-sensitive robustness, $\mathrm{Rob_{non}}$ is certified robustness of non-sensitive samples. The best statistics are highlighted in bold.

| Dataset | Type | Methods | Acc | $\mathrm{Rob_{c\text{-}s}}$ | $\mathrm{Rob_{non}}$ |
|---|---|---|---|---|---|
| CIFAR-10 | Single ($3 \to 5$) | Cohen | 0.654 | 0.504 | 0.443 |
| | | MACER | 0.647 | 0.543 | 0.533 |
| | | Cohen-R | 0.642 | 0.723 | 0.437 |
| | | Ours | **0.673** | **0.924** | 0.472 |
| | Multi ($3 \to 2, 4, 5$) | Cohen | **0.654** | 0.336 | 0.446 |
| | | MACER | 0.647 | 0.385 | 0.533 |
| | | Cohen-R | 0.642 | 0.643 | 0.437 |
| | | Ours | 0.643 | **0.822** | 0.474 |
| Imagenette | Single ($7 \to 2$) | Cohen | **0.803** | 0.885 | 0.744 |
| | | MACER | 0.782 | 0.899 | 0.749 |
| | | Cohen-R | 0.754 | 0.911 | 0.679 |
| | | Ours | 0.792 | **0.938** | 0.731 |
| | Multi ($7 \to 2, 4, 6$) | Cohen | **0.803** | 0.756 | 0.744 |
| | | MACER | 0.781 | 0.780 | 0.749 |
| | | Cohen-R | 0.754 | 0.830 | 0.678 |
| | | Ours | 0.796 | **0.863** | 0.730 |

$\gamma$. MACER uses $\gamma = 8$ to enhance the overall robustness for all classes, whereas we set $\gamma_1 = 4$ for normal classes and $\gamma_2 = 16$ for sensitive classes based on our analysis in Section 4.2. The effect of different combinations of hyperparameters is discussed in Appendix E. In addition, we compare our method with an alternative convex relaxation-based approach (Zhang & Evans, 2019) in Appendix D.

## 5.1 CERTIFICATION RESULTS FOR DIFFERENT COST MATRICES

**Seedwise Cost Matrix.** For any $(\boldsymbol{x}, y) \in \mathcal{S}^s$, $\Omega_y = \{j \in [m] : j \neq y\}$ or $|\Omega_y| = m - 1$, meaning that any possible classwise adversarial transformation for $(\boldsymbol{x}, y)$ will incur a cost. Table 1 reports the performance in terms of overall accuracy and certified cost-sensitive robustness of our method and the three baselines with respect to different seedwise cost matrices. In particular, we consider two types of seedwise cost matrices in our experiments: (2) *Single:* a randomly-selected sensitive seed class from all available classes, where we report the performance on the third class "cat" (label 3) for CIFAR-10 and "gas pump"(label 7) for Imagenette. (3) *Multi:* multiple sensitive seed classes, where "bird" (label 2) and "deer" (label 4) are considered as the sensitive seed classes for CIFAR-10, while we choose "chain saw"(label 3) and "gas pump"(label 7) for Imagenette.

We observe in Table 1 that our cost-sensitive robust training method achieves a significant improvement in terms of certified cost-sensitive robustness compared with baselines. In certain cost-sensitive scenarios, our method achieves comparable or even slightly better performance than the baselines in terms of overall accuracy. The certified robustness of our method for non-sensitive examples is not as high as MACER's, indicating a desirable shift of robustness from normal to sensitive examples.

**Pairwise Cost Matrix.** For any $(\boldsymbol{x}, y) \in \mathcal{S}^s$, $\Omega_y \subseteq \{j \in [m] : j \neq y\}$ with $|\Omega_y| < m - 1$, meaning that misclassification to any target class in $[m] \backslash \Omega_y$ is acceptable. Note that $[m] \backslash \Omega_y$ may include target classes other than the ground-truth class $y$. Similar to the previous setting, we consider two types of pairwise cost matrices: (1) *Single:* a randomly-selected sensitive seed class with one sensitive target class. (2) *Multi:* a single sensitive seed class with multiple sensitive target classes. Table 2 compares the performance of our method with baselines for the aforementioned pairwise cost matrices on CIFAR-10 and Imagenette datasets. Similar to results in seedwise cost matrix, our methods always yeilds the optimal cost-sensitive certified robustness without sacrificing too much overall accuracy. The superiority of our approach in preserving overall accuracy becomes even more pronounced when contrasted with the reweighting method *Cohen-R*.

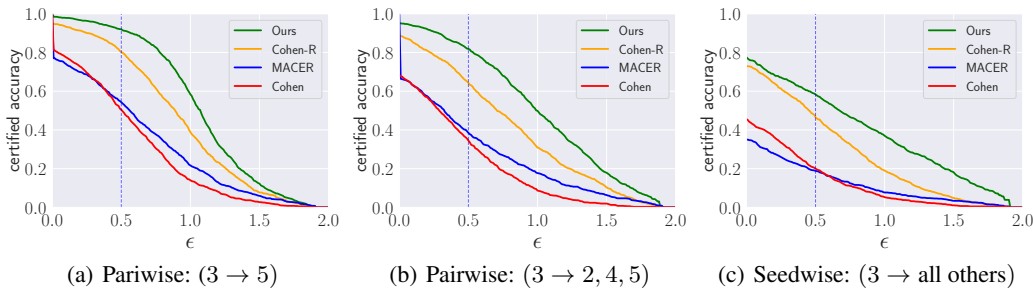

Figure 2: The curves of certified accuracy with respect to different training methods on the CIFAR-10 dataset for varying $\epsilon$ under different cost-sensitive settings.

## 5.2 MEDICAL DATASET

We report the results of our method on a real-world medical dataset: HAM10k, which is an imbalanced skin cancer dataset with 7 classes. The noise level here is set as $\sigma = 0.5$ and the cost-sensitive robustness $\text{Rob}_{\text{c-s}}$ is measured for $\epsilon = 0.5$ in $\ell_2$-norm. Due to the small sample size for several classes, we group all images from the benign classes into a single category and all images from the malignant classes into another, which formulates a binary classification task. Since misclassifying a malignant tumor as benign can have severe consequences and resulting in high costs for this medical application, we set any misclassiciation from malignant to benign as cost-sensitive, while regarding the cost of the other type of misclassifications as 0. Table 3 demonstrates the overall accuracy and certified cost-

Table 3: Comparisons of different methods on HAM10k dataset.

| Method | Acc | $\text{Rob}_{\text{c-s}}$ |
|---|---|---|
| Cohen | 0.829 | 0.118 |
| Cohen-R-I | 0.805 | 0.197 |
| Cohen-R-II | 0.784 | **0.437** |
| MACER | 0.828 | 0.250 |
| Ours | **0.831** | 0.413 |

sensitive robustness of the model produced by our cost-sensitive robust training method, compared with other alternatives. Cohen-R-I and Cohen-R-II represent two specific instantiations of Cohen-R with the weight parameter $\alpha$ selected as 1.1 and 1.2, respectively. As we increase the weight parameter $\alpha$, there is a noticeable enhancement in cost-sensitive robustness. However, this comes at the expense of a marked decline in overall accuracy, confirming the non-optimal trade-off phenomenon discussed in Section 4. Notably, our approach achieves a more desirable trade-off between the two metrics, underscoring its effectiveness for real-world applications.

## 5.3 CERTIFICATION RESULTS UNDER VARYING $\epsilon$

To further visualize the consistency of our improvement, we compare the certified accuracy curves of cost-sensitive examples with varying $l_2$ perturbation for the aforementioned methods in Figure 2. It is evident that our method consistently outperforms the baseline in terms of certified cost-sensitive accuracy across different $\epsilon$. In particular, we observe a significant improvement at $\epsilon = 0.5$ of our method compared with other baselines, which again confirms our findings in Section 5.1.

## 6 CONCLUSION

We developed a generic randomized smoothing framework to certify and train for cost-sensitive robustness. At the core of our framework is a new notion of cost-sensitive certified radius, which is applicable to any binary cost matrix. Built upon fine-grained thresholding techniques for optimizing the certified radius with respect to different subpopulations, our method significantly improves the certified robustness performance for cost-sensitive transformations. Compared with naive reweighting approaches, our method achieves a much more desirable trade-off between overall accuracy and certified cost-sensitive robustness. Experiments on image benchmarks demonstrate the superior performance of our approach compared to various baselines. Our work opens up new possibilities for building certified robust models based on randomized smoothing for cost-sensitive applications.

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

# A    RELATED WORK

Cohen et al. (2019) proposed randomized smoothing, which first converts any base neural network into a smoothed classifier by injecting Gaussian noises to inputs followed by majority voting, then provides a robust certificate that can guarantee the prediction of the resulting smoothed classifier is constant within some $\ell_2$-norm ball for any given input. Compared with other certification methods, a biggest advantage of the randomized smoothing framework is its scalability to large neural networks and large-scale datasets. In particular, Cohen et al. (2019) provided a tight robustness guarantee for randomized smoothing with $\ell_2$ perturbations. Later on, SmoothADV (Shafahi et al., 2019) improved the proposed training method in Cohen et al. (2019) by designing an adaptive attack on the smoothed classifier using adversarial training and first-order approximations. In addition, MACER (Zhai et al., 2020) developed a more direct way which directly optimizes the smoothed classifiers' certified radius with respect to correctly-classified samples using margin based loss and achieves better robustness and accuracy trade-off than previous methods. More recently, several research studies (Carlini et al., 2022; Xiao et al., 2022; Zhang et al., 2023) proposed to improve the base classifier training pipeline by adapting pretrained diffusion models to denoise the Gaussian augmented samples.

Cost-sensitive learning deals with the situation where different misclassifications will induce different costs (Domingos, 1999; Elkan, 2001). For example, misclassifying a malicious tumor to benign (Khan et al., 2017) will bring more harmful consequences to the patient than the reverse. For empirical methods such as adversarial training, it is also valuable to make the classifier adapt to cost-sensitive settings so that adversarial transformations with high costs will be less likely to happen. Most of the cost-sensitive robust training methods are applicable to either simple linear classifiers or empirical training methods that cannot produce robust certificate (Khan et al., 2017; Chen et al., 2021). Zhang & Evans (2019) proposed a method to train cost-sensitive certifiable classifiers using certified methods based on convex optimization, however, it can not scale to large neural network or large datasets such as ImageNet. Our work combines randomized smoothing and cost-sensitive learning to provide more scalable classifiers with good certifiable robustness guarantees under cost-sensitive scenarios.

# B    COMPARISONS WITH STANDARD CERTIFIED RADIUS

The main distinction between our *cost-sensitive certified radius* $R_{\text{c-s}}(\boldsymbol{x}; \Omega_y)$ and Cohen et al. (2019)'s standard certified radius $R(\boldsymbol{x})$ lies in the case when $|\Omega_y| < m - 1$. In the following, we provide the detailed proof of Theorem 3.3.

*Proof of Theorem 3.3.*  Recall Cohen's certified radius is:

$$R(\boldsymbol{x}) = \frac{\sigma}{2}\Big[\Phi^{-1}\big([h_\theta(\boldsymbol{x})]_y\big) - \Phi^{-1}\big(\max_{k \neq y}[h_\theta(\boldsymbol{x})]_k\big)\Big], \tag{5}$$

note the prerequisite in the definition of $R(\boldsymbol{x})$ guarantees that $\boldsymbol{x}$ is correctly classified, which means $y$ is both the ground-truth class and top-1 class. our *cost-sensitive certified radius* is:

$$R_{\text{c-s}}(\boldsymbol{x}; \Omega_y) = \frac{\sigma}{2}\Big[\Phi^{-1}\big(\max_{k \in [m]}\big[h_\theta(\boldsymbol{x})\big]_k\big) - \Phi^{-1}\big(\max_{k \in \Omega_y}\big[h_\theta(\boldsymbol{x})\big]_k\big)\Big],$$

If $|\Omega_y| = m - 1$, $R_{\text{c-s}}(\boldsymbol{x}; \Omega_y)$ and $R(\boldsymbol{x})$ are equivalent as the set of target classes in both definitions are identical, while if $|\Omega_y| < m - 1$, the second term in *cost-sensitive certified radius* will be no bigger than that of Cohen's certified radius, leading to $R_{\text{c-s}}(\boldsymbol{x}; \Omega_y) \geq R(\boldsymbol{x})$.

To be more specific, we have the following observations:

1. When $|\Omega_y| = m - 1$, $R_{\text{c-s}}(\boldsymbol{x}; \Omega_y) \Leftrightarrow R(\boldsymbol{x})$. Since $\Omega_y = \{j | j \neq y, j \in [m]\}$ encompasses all incorrect classes, the two probability terms are fully matched for both radius.

2. When $|\Omega_y| < m - 1$, $R_{\text{c-s}}(\boldsymbol{x}; \Omega_y) \geq R(\boldsymbol{x})$. This is because the class index scope in the second term of $R_{\text{c-s}}(\boldsymbol{x}; \Omega_y)$ is narrower than that in $R(\boldsymbol{x})$. Since $\Omega_y \subseteq \{j \neq y, j \in [m]\}$, the highest probability w.r.t $|\Omega_y|$ will be no bigger than the highest probability related to all incorrect classes in $R(\boldsymbol{x})$. Consequently, $R_{\text{c-s}}(\boldsymbol{x}; \Omega_y) \geq R(\boldsymbol{x})$.

Therefore, we complete the proof of Theorem 3.3.    □

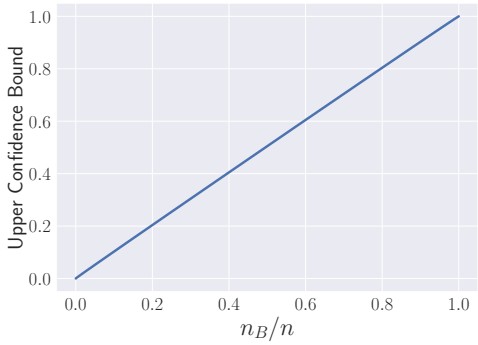

Figure 3: Illustration of the relationship between the top-class sampling ratio $\hat{p}_B = n_B/n$ and its corresponding probabilistic upper confidence bound. Here, $n_B$ represents the number of samples that is predicted as the top class in $\Omega_y$, and $n$ represents total sampling number.

## C  ADDITIONAL DETAILS FOR SECTION 3.2

This section provides more details about Algorithm 1 and the proof of Theorem 3.4.

**Details of Algorithm 1.** We follow the same sampling procedure of Cohen et al. (Cohen et al., 2019). To be more specific, the sampling function SAMPLEUNDERNOISE($f, x, n, \sigma$) is defined as:

1. Draw $n$ i.i.d. samples of Gaussian noises $\boldsymbol{\delta}_1 \ldots \boldsymbol{\delta}_n \sim \mathcal{N}(0, \sigma^2 \mathbf{I})$.
2. Obtain the predictions $f(\boldsymbol{x} + \boldsymbol{\delta}_1), \ldots, f(\boldsymbol{x} + \boldsymbol{\delta}_n)$ with base classifier $f$ on noisy images.
3. Return the counts for each class, where the count for class c is $\sum_{i \in [n]} \mathbb{1}[f(\boldsymbol{x} + \boldsymbol{\delta}_i) = c]$.

**Proof of Theorem 3.4.** We provide the proof of Theorem 3.4 presented in Section 3.2, where we make use of the union bound to prove the validity the upper confidence bound $\overline{p_B}$ with respect to $\Omega_y$.

*Proof of Theorem 3.4.* Our goal is to show that $R_2$ specified in Algorithm 1 is a cost-sensitive certified radius with at least $1 - \alpha$ confidence over the randomness of the Gaussian sampling. Let $m$ be the number of total label classes, and let $(p_1, \ldots, p_m)$ be the ground-truth probability distribution of the smoothed classifier $g_\theta$ for a given example $(\boldsymbol{x}, y)$. Denote by $p_A, p_B$ the maximum probabilities in $[m]$ and in $\Omega_y$, respectively. According to the design of Algorithm 1, we can compute the empirical estimate of $p_k$ for any $k \in [m]$ based on $n$ Gaussian samples and the base classifier $f_\theta$. Let $(\hat{p}_1, \ldots, \hat{p}_m)$ be the corresponding empirical estimates, then we immediately know

$$\hat{p}_k \sim \text{Binomial}(n, p_k), \text{for any } k \in [m].$$

For $R_1$, we follow the same procedure as in Cohen et al. (2019) to compute the $1 - \alpha$ lower confidence bound on $p_A$, which is guaranteed by the fact that $p_B \le 1 - p_A$. However, for the computation of $R_2$, we need compute both a lower confidence bound on $p_A$ and a upper confidence bound on $p_B$, which requires additional care to make the computation rigorous. In particular, we adapt the definition of standard certified radius (Theorem 1 in Cohen et al. (2019)) to cost-sensitive scenarios for deriving $R_2$. According to the construction of $\underline{p_A} = \text{LowerConfBnd}(\text{count}[\hat{c}_A], n, 1 - \alpha/2)$, we have

$$\Pr\left[\underline{p_A} \le p_A\right] \ge 1 - \frac{\alpha}{2}.$$

Therefore, the remaining task is to prove

$$\Pr\left[\overline{p_B} \ge p_B\right] \ge 1 - \frac{\alpha}{2}, \tag{6}$$

where $\overline{p_B} = \text{UpperConfBnd}(\text{count}[\hat{c}_B], n, 1 - \alpha/(2|\Omega_y|))$, and $\Omega_y$ denotes the set of cost-sensitive target classes. Note that $p_B = \max_{k \in \Omega_y}\{p_k\}$ is used to define $R_{\text{c-s}}(\boldsymbol{x}, \Omega_y)$. If the above inequality

holds true, we immediately know that by union bound,

$$\Pr\Big[R_2 \leq R_{\text{c-s}}(\boldsymbol{x}; \Omega_y)\Big] = \Pr\Big[\frac{\sigma}{2}\big(\Phi^{-1}(\underline{p_A}) - \Phi^{-1}(\overline{p_B})\big) \leq \frac{\sigma}{2}\big(\Phi^{-1}(p_A) - \Phi^{-1}(p_B)\big)\Big]$$

$$\geq 1 - \Big(\Pr\Big[\underline{p_A} \geq p_A\Big] + \Pr\Big[\overline{p_B} \leq p_B\Big]\Big) \geq 1 - \alpha.$$

The challenge for proving Equation 6 lies in the that we do not know the top class within $\Omega_y$ which is different from the case of $p_A$. Therefore, we resort to upper bound the maximum over all the ground-truth class probabilities within $\Omega_y$. Based on the distribution of $\hat{p}_k$, we know for any $k \in \Omega_y$,

$$\Pr\Big[\overline{p_k} \geq p_k\Big] \geq 1 - \alpha/(2|\Omega_y|),$$

where $\overline{p_k}$ is defined as the $1 - \alpha/(2|\Omega_y|)$ upper confidence bound using $\hat{p}_k$ similar to the construction of $\overline{p_B}$. We remark that the choice of $1 - \alpha/(2|\Omega_y|)$ can in fact be varied for each $k \in \Omega_y$ and even optimized for obtaining tighter bounds, as long as the summation of the probabilities of bad event happening is at most $\alpha/2$. We set $1 - \alpha/(2|\Omega_y|)$ to be the same accross different $k$ for simplicity. According to union bound, we have

$$\Pr\Big[\max_{k \in \Omega_y} \overline{p_k} \geq p_B\Big] = \Pr\Big[\max_{k \in \Omega_y} \overline{p_k} \geq \max_{k \in \Omega_y} p_k\Big]$$

$$\geq 1 - \sum_{k \in \Omega_y} \Pr\Big[\overline{p_k} \leq p_k\Big]$$

$$\geq 1 - |\Omega_y| \cdot \alpha/(2|\Omega_y|) = 1 - \frac{\alpha}{2},$$

where the first inequality holds because of union bound. Finally, since the upper confidence bound is monotonically increasing with respect to the value of $\hat{p}$ within $[0, 1]$ (see Figure 3 for a visualization of such relationship), we know $\overline{p_B} = \text{UpperConfBnd}(\text{count}[\hat{c}_B], n, 1 - \alpha/(2|\Omega_y|))$ will the be largest upper bound if $\text{count}[\hat{c}_B]$ is the maximum. Therefore, we complete the proof. □

## D    COMPARISONS WITH ZHANG & EVANS (2019)

Zhang & Evans (2019) proposed a method to certify cost-sensitive robustness of any classifier based on convex relaxation  (Wong & Kolter, 2018), which provides a robustness guarantee for a given input via minimizing the worst-case loss within the relaxed convex outer polytype. Also, Zhang & Evans (2019) developed a training method for training provably cost-sensitive robust classifiers. In particular, their method incorporates different types of cost matrices into the convex optimization process to train cost-sensitive robust classifiers. However, the initial work of Wong & Kolter (2018) only focuses on $\ell_\infty$-norm bounded perturbations and does not consider perturbations in $\ell_2$-norm. As a result, the proposed method in  Zhang & Evans (2019) also did not address the cost-sensitive robustness for $\ell_2$ perturbations. We note that in a follow-up work of Wong et al. (2018), they extend the developed certification techinques to $\ell_2$ perturbations. For fair comparisons with our method, we further extend the cost-senstive robust learning method of Zhang & Evans (2019) to handle $\ell_2$-norm perturbations using the method of Wong et al. (2018). We report their comparisons in Table 4, the certified cost-sensitive robustness for the convex-relaxation method is computed as the *cost-sensitive robust error* defined in Zhang & Evans (2019), which represents the fraction of test samples that are guaranteed to be robust to certain $\ell_2$ perturbations.

Table 4: Comparisons of our method with convex relaxation based method (Zhang & Evans, 2019) for $\ell_2$-norm on CIFAR-10, where a single pairwise transformation ($3 \rightarrow 5$) is considered sensitive.

| Method | $\ell_2$ Perturbations | Cost-Sensitive Robustness | Overall Accuracy |
|---|---|---|---|
| Zhang & Evans (2019) | $\epsilon = 0.25$ | 0.944 | 0.480 |
| Ours | $\epsilon = 0.5$ | 0.924 | 0.673 |

Table 4 shows that our method achieves much higher overall accuracy even against larger $\ell_2$ perturbations, suggesting a better cost-sensitive robustness and overall accuracy trade-off. Also, we find in our

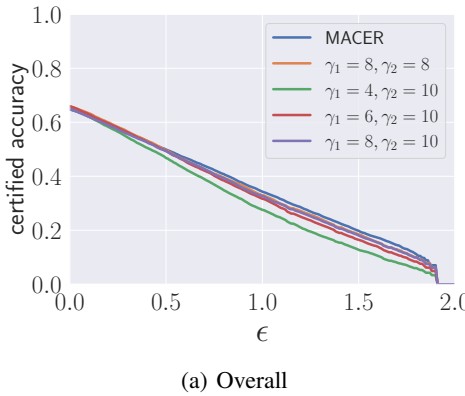 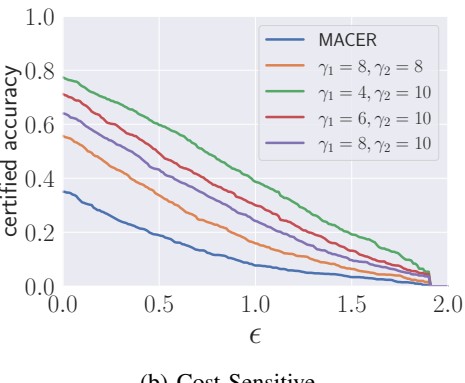

(a) Overall          (b) Cost-Sensitive

Figure 4: Visualizations of our method with $\gamma_1 \in \{4, 6, 8\}$ and fixed $\gamma_2 = 10$ with comparisons to baseline methods with: (a) overall performance and (b) cost-sensitive performance. Here, we consider a single cost-sensitive seed class "Cat" for the cost matrix.

implementation that convex relaxation-based methods is not applicable to large $\ell_2$ perturbations (e.g., $\epsilon = 0.5$), due to memory issues. We also remark that randomized smoothing techniques proposed in existing works (Cohen et al., 2019; Li et al., 2019; Jia et al., 2019) primarily focus on defending against $\ell_2$-perturbations. As a result, our methods excel in achieving good cost-sensitive performance under $\ell_2$-norm bounded perturbations. There are limitations when it comes to certifying cost-sensitive robustness using our method under other types of perturbations, such as perturbations with $\ell_1$-norm, $\ell_\infty$-norm and even beyond $\ell_p$-norm.

## E    HYPERPARAMETER TUNING

In this section, we study the effect of hyperparameters $\gamma_1$ and $\gamma_2$ used in our method proposed in Section 4.2 on the two evaluation metrics, certified overall accuracy and cost-sensitive robustness. Note that our goal is to improve cost-sensitive robustness without sacrificing overall accuracy, where $\gamma_1$ controls the margin of normal classes and $\gamma_2$ controls the margin of sensitive classes. In particular, we report the parameter tuning results on CIFAR-10. Here, the cost matrix is selected as a seedwise cost matrix with a sensitive seed class "cat". We choose the specific "cat" class only for the purpose of illustration, as we observe similar trends in our experiments for other cost matrices, similar to the results shown in Table 1.

In addition, we consider two comparison baselines:

1. MACER (Zhai et al., 2020) with $\gamma = 8$, restricting only on correctly classified examples.

2. Our method with $\gamma_1 = 8$ and $\gamma_2 = 8$, the only difference with MACER is that our method contains misclassified examples for sensitive classes.

Below, we show the effect of $\gamma_1$ and $\gamma_2$ on the performance of our method, respectively.

**Effect of $\gamma_1$.** Note that $\gamma_1$ is used to restrict the certified radius with respect to normal data points. Figure 4 illustrates the influence of varying $\gamma_1 \in \{4, 6, 8\}$ and fixed $\gamma_2 = 10$ for our method, with comparisons to the two baselines, on both overall accuracy and cost-sensitive robustness.

For the original implementation of MACER, $\gamma$ is selected as 8 for the best overall performance. Although it achieves good overall robustness, it does not work for cost-sensitive settings, which suggests the possibility of a trade-off space, where different classes can be balanced to achieve our desired goal of cost-sensitive robustness. The second baseline is our method with $\gamma_1 = 8$ and $\gamma_2 = 8$. By incorporating misclassified samples for sensitive seed class, the cost-sensitive performance substantially improvemes. This results shows the significance of including misclassified sensitive samples during the optimization process of the certified radius.

Table 5: Performance of our method for different combinations of $\gamma_1$ and $\gamma_2$. Here, the cost-sensitive scenario is captured by the seedwise cost matrix with a single sensitive seed class "Cat" for CIFAR-10.

| Method | $\gamma_1$ | $\gamma_2$ | Acc | $\text{Rob}_{\text{c-s}}$ |
|---|---|---|---|---|
| MACER | - | - | 0.647 | 0.189 |
| Ours | 8 | 8 | 0.660 | 0.338 |
| Ours | 2 | 8 | 0.654 | 0.633 |
| | 2 | 10 | 0.634 | 0.687 |
| | 2 | 12 | 0.637 | 0.691 |
| | 2 | 16 | 0.630 | 0.705 |
| Ours | 4 | 8 | 0.670 | 0.507 |
| | 4 | 10 | 0.653 | 0.597 |
| | 4 | 12 | 0.659 | 0.576 |
| | 4 | 16 | 0.661 | 0.583 |
| Ours | 6 | 8 | 0.673 | 0.396 |
| | 6 | 10 | 0.660 | 0.493 |
| | 6 | 12 | 0.655 | 0.544 |
| | 6 | 16 | 0.649 | 0.552 |
| Ours | 8 | 8 | 0.660 | 0.338 |
| | 8 | 10 | 0.650 | 0.432 |
| | 8 | 12 | 0.641 | 0.474 |
| | 8 | 16 | 0.645 | 0.463 |

Moreover, we can observe from Figure 4(b) that as we reduce the value of $\gamma_1$, the robustness performance of the cost-sensitive seed class increases. This again confirms that by limiting the certified radius of normal classes to a small threshold in our method, the model can prioritize sensitive classes and enhance cost-sensitive robustness.

**Effect of $\gamma_2$.** Figure 5 illustrates the influence of varying $\gamma_2 \in \{8, 12, 16\}$ with fixed $\gamma_1 = 4$ or fixed $\gamma_1 = 8$ for our method, with comparisons to the two baselines, on both overall accuracy and cost-sensitive robustness. Moreover, we can observe from Figure 5(b) and Figure 5(d) that as we increase the value of $\gamma_2$, the robustness performance of the cost-sensitive seed class increases. This confirms that by optimizing the certified radius of sensitive classes to a large threshold in our method, the model can focus more on sensitive classes and enhance cost-sensitive robustness. Additionally, there is a slight increase in the overall certified accuracy. This can be attributed to the fact that the overall accuracy takes into account both the accuracy of sensitive samples and normal samples. As the certified accuracy of sensitive samples increases, it dominates the overall accuracy and leads to its overall improvement. Table 5 demonstrates the impact of different combinations of hyperparameters of $(\gamma_1, \gamma_2)$ on both the overall accuracy and cost-sensitive performance. The choice of $\gamma_1$ and $\gamma_2$ is crucial and requires careful consideration. For $\gamma_2$, setting a value that is too small can greatly undermine the overall accuracy, even though it may improve cost-sensitive robustness. This is because the performance of normal classes deteriorates, resulting in a degradation of overall performance. Otherwise, if the value is too large (i.e., $\gamma_2 = 8$), it has a negative impact on cost-sensitive performance.

Regarding $\gamma_1$, it is evident that increasing its value while keeping $\gamma_2$ fixed leads to a significant improvement in cost-sensitive robustness. It is worth noting that even though the cost-sensitive seed class represents only a single seed, accounting for only $10\%$ of the total classes, enhancing its robustness has a positive effect on overall accuracy as well. For instance, let's compare the combination $(\gamma_1 = 8, \gamma_2 = 4)$ to $(\gamma_1 = 8, \gamma_2 = 8)$. We observe that the former, which exhibits better cost-sensitive robustness, outperforms the latter in terms of both overall accuracy and cost-sensitive robustness. It achieves an approximate improvement of $1.52\%$ in overall accuracy and a significant improvement of approximately $50\%$ in cost-sensitive robustness. This finding highlights the effectiveness of our sub-population-based methods. It demonstrates that by fine-tuning the

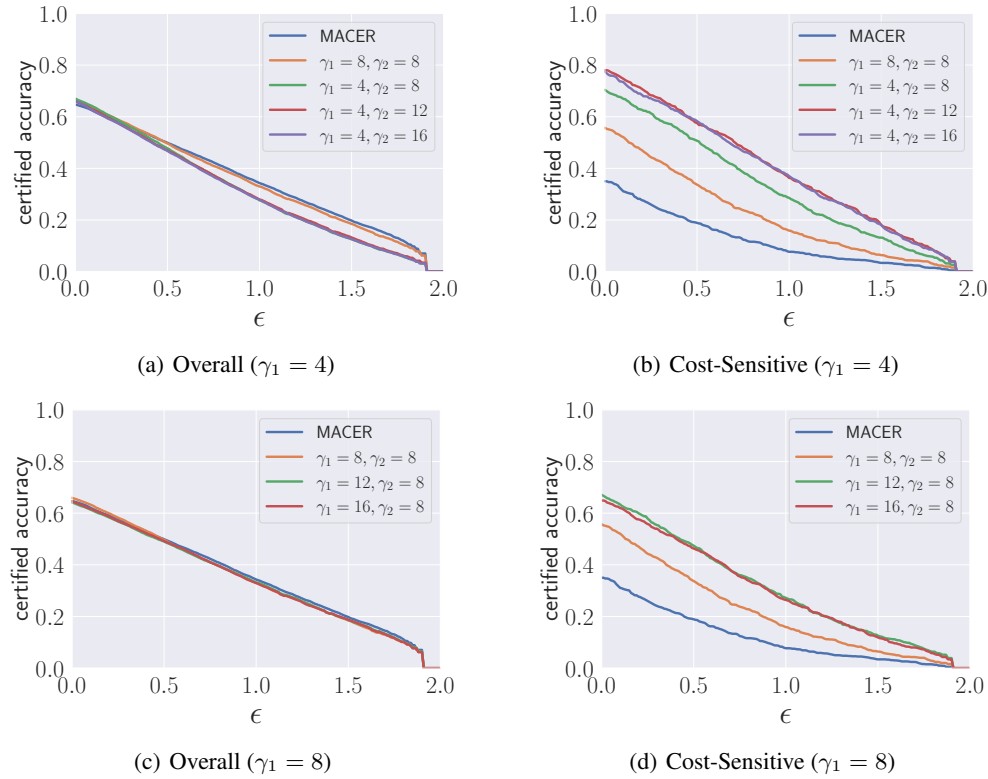

Figure 5: Visualizations of our method for two groups comparisons to baseline methods in terms of (a)(c) overall performance and (b)(d) cost-sensitive performance. The first with $\gamma_2 \in \{8, 12, 16\}$ and fixed $\gamma_1 = 4$, the second with $\gamma_2 \in \{8, 12, 16\}$ and fixed $\gamma_1 = 8$. The cost matrix is set as the matrix representing a single cost-sensitive seed class "Cat".

optimization thresholds for the certified radius of sensitive classes and normal classes separately, we can achieve a better trade-off between overall accuracy and cost-sensitive robustness.

## F VARYING NOISE

We also tested our method's performance when the injecting noise $\sigma = 0.25$. The results are demonstrated in Table 6 and Table 7 for seedwise and pairwise cost matrices, which show that our method consistently outperforms the two baseline randomized smoothing methods and the reweighting method of Cohen-R.

Table 7: Certification results for pairwise cost matrices. The noise level $\sigma$ is 0.25 for both CIFAR-10 and Imagenette. Acc stands for overall accuracy and $\text{Rob}_{\text{c-s}}$ refers to certified cost-sensitive robustness with $\epsilon = 0.25$, $\text{Rob}_{\text{non}}$ denotes the certified robustness of non-sensitive samples with $\epsilon = 0.25$. The best results are highlighted in bold.

| Dataset | Type | Method | Acc | $\text{Rob}_{\text{c-s}}$ | $\text{Rob}_{\text{non}}$ |
|---|---|---|---|---|---|
| CIFAR-10 | Single ($3 \to 5$) | Cohen | 0.793 | 0.674 | 0.679 |
| | | MACER | 0.808 | 0.709 | 0.730 |
| | | Cohen-R | 0.778 | 0.811 | 0.642 |
| | | Ours | **0.809** | **0.836** | 0.691 |
| | Multi ($3 \to 2, 4, 5$) | Cohen | 0.793 | 0.515 | 0.679 |
| | | MACER | 0.808 | 0.582 | 0.785 |
| | | Cohen-R | 0.778 | 0.685 | 0.642 |
| | | Ours | **0.807** | **0.689** | 0.692 |
| Imagenette | Single ($7 \to 2$) | Cohen | 0.803 | 0.880 | 0.775 |
| | | MACER | 0.796 | 0.837 | 0.786 |
| | | Cohen-R | 0.764 | 0.921 | 0.691 |
| | | Ours | **0.830** | **0.942** | 0.809 |
| | Multi ($7 \to 2, 4, 6$) | Cohen | 0.803 | 0.730 | 0.775 |
| | | MACER | 0.796 | 0.780 | 0.786 |
| | | Cohen-R | 0.764 | 0.831 | 0.691 |
| | | Ours | **0.832** | **0.842** | 0.799 |

Table 6: Certification results of different randomized smoothing based training methods for various seedwise cost matrices. The noise level $\sigma$ is 0.25 for both CIFAR-10 and Imagenette. Acc stands for overall accuracy, $\text{Rob}_{\text{c-s}}$ is certified cost-sensitive robustness with $\epsilon = 0.25$, $\text{Rob}_{\text{non}}$ is certified robustness of non-sensitive samples with $\epsilon = 0.25$. Best statistics are highlighted in bold.

| Dataset | Type | Method | Acc | $\text{Rob}_{\text{c-s}}$ | $\text{Rob}_{\text{std}}$ | $\text{Rob}_{\text{non}}$ |
|---|---|---|---|---|---|---|
| CIFAR-10 | Single (3) | Cohen | 0.793 | 0.407 | 0.387 | 0.679 |
| | | MACER | 0.808 | 0.475 | 0.454 | 0.730 |
| | | Cohen-R | 0.778 | 0.583 | 0.568 | 0.642 |
| | | Ours | **0.804** | **0.602** | 0.570 | 0.684 |
| | Multi (2, 4) | Cohen | 0.793 | 0.588 | 0.556 | 0.728 |
| | | MACER | **0.808** | 0.653 | 0.629 | 0.785 |
| | | Cohen-R | 0.784 | 0.635 | 0.605 | 0.702 |
| | | Ours | 0.807 | **0.717** | 0.685 | 0.719 |
| Imagenette | Single (7) | Cohen | 0.803 | 0.656 | 0.656 | 0.775 |
| | | MACER | 0.796 | 0.701 | 0.699 | 0.786 |
| | | Cohen-R | 0.738 | 0.742 | 0.742 | 0.691 |
| | | Ours | **0.836** | **0.756** | 0.756 | 0.799 |
| | Multi (3, 7) | Cohen | 0.803 | 0.612 | 0.612 | 0.801 |
| | | MACER | 0.796 | 0.662 | 0.653 | 0.777 |
| | | Cohen-R | 0.764 | 0.656 | 0.656 | 0.736 |
| | | Ours | **0.830** | **0.702** | 0.702 | 0.802 |

