# OpenReview forum: "Provably Robust Cost-Sensitive Learning via Randomized Smoothing"
_ICLR.cc/2024/Conference — ICLR 2024 Conference Withdrawn Submission_

### Official Review · Reviewer_F4hf · 2023-10-26

**Soundness:** 3 good
**Presentation:** 3 good
**Contribution:** 2 fair
**Rating:** 3
**Confidence:** 5

**Summary:**

This paper introduces randomized smoothing to cost-sensitive learning problems to develop a provable robust cost-sensitive learning framework. Empirical studies are conducted to show the effectiveness of the proposed method.

**Strengths:**

Cost-sensitive learning is an essential task in the ML community, and considering how to obtain a robust model under such a context is meaningful.

**Weaknesses:**

The contributions of this paper are somewhat weak since some conclusions claimed in the main paper could be directly realized by Cohen et al. (2019). Furthermore, in Alg.1, where the $R_1$ comes from? In Cohen's paper, they also considered this in a binary classification case.

Another non-neglected limitation of this paper is that it can only process the binary cost-matrix problem. This is also a possible reason why the Cohen-R algorithm cannot perform well.

Also, the motivation of Sec.4.2 needs to be further added.
- Why can Eq.(4) work well? Why must we do the optimization like this?
- I noticed that $I_2$ also includes the cost-sensitive sample. But why the cost-sensitive samples included in $I_2$ is processed differently from I_3?
- Moreover, the certified radii R_{c-s} would be greater than $0$ since the set $[m]$ includes $\Omega_y$. Thus, why does $I_3$ exist $-\gamma_2$？
- Generally speaking, we usually do not obtain the exact value of g due to the expectation. Thus, in Eq.(4), how do you back-propagate $g(x)$?

Finally, the experiment part is somewhat unconvincing. The authors should consider more of the latest randomizing smooth methods, such as [1,2].

Ref:
- [1] Consistency regularization for certified robustness of smoothed classifiers, NeurIPS, 2020.
- [2] Provably robust deep learning via adversarially trained smoothed classifiers, NeurIPS, 2019.

**Questions:**

See the Weakness part above.

---

### Official Review · Reviewer_i4At · 2023-10-28

**Soundness:** 3 good
**Presentation:** 3 good
**Contribution:** 1 poor
**Rating:** 3
**Confidence:** 4

**Summary:**

This paper studies provable adversarial robustness in the context of cost-sensitive learning. It applies randomized smoothing to obtain a certified radius where the classification cost is zero. In the cost-sensitive setting, a misclassification incurs a cost defined by an m x m matrix C, where m is the number of class labels. The entry C_{i, j} denotes the cost of misclassifying label i as j. The paper studies a special case of this matrix where all entries are 0 or 1. The traditional classification setting, where the misclassification cost is uniform, is captured by a matrix where all diagonal elements are 0, and all non-diagonal elements are 1. In the more general case, some off-diagonal entries could be 0, indicating no misclassification cost for the corresponding class. This offers the potential to increase the certified radius by permitting certain misclassifications, given that their associated cost is nil.

**Strengths:**

The paper studies provable robustness in a setting that has largely remained unexplored. The paper is well-written and easy to follow.

**Weaknesses:**

1. Binary Cost Matrix: A 0/1 cost matrix may not be sufficient to capture real-world cost-sensitive ML tasks. The cost of misclassification would rarely be zero. Take the example mentioned in the second paragraph of the introduction. While misclassifying a benign tumor as malignant is less detrimental than the reverse, the cost of such a misclassification will not be zero. If so, one could simply label all tumors as malignant and achieve an overall classification cost of zero. However, such a classifier would not provide us with any valuable information.

   It would be more impactful to design robustness certificates for a general cost matrix where the entries need not be 0/1. It might be possible to certify the expected misclassification cost by using the distribution of the cost under the smoothing noise. The following work on certifying the expected confidence of a neural network could be adapted for a general cost matrix:

   Certifying Confidence via Randomized Smoothing, Kumar et al., NeurIPS 2020.

2. Novelty: The robustness certificate designed in this work is a straightforward adaptation of Cohen et al.'s certificate [1]. The certificate in [1] takes the difference between two terms where the second term depends on p_b, the probability of the second most likely class. This paper redefines p_b as the probability of the most likely class in the set of class labels with cost 1.

   [1] Certified Adversarial Robustness via Randomized Smoothing, Cohen et al., ICML 2019.

3. Sample Complexity: The number of samples required for computing the proposed certificate is higher than that of the baseline certificate from [1]. It depends on the number of classes with cost 1. While this might be manageable for the small number of classes considered in the experiments (<= 10), scaling to a large number of classes, such as in ImageNet (1000 classes), would be difficult.

**Questions:**

1. What would be a practical application where a binary cost matrix would be sufficient? In most scenarios, the misclassification cost would take a range of different values.

2. During inference, when the ground truth is unknown, how do we find the classes with cost 1? This set depends on the correct class label, which is not known during inference.

---

### Official Review · Reviewer_oovB · 2023-10-31

**Soundness:** 3 good
**Presentation:** 3 good
**Contribution:** 2 fair
**Rating:** 6
**Confidence:** 3

**Summary:**

This work proposed randomized smoothing certification for cost-sensitive learning, where misclassifying one class as another has different costs. The work provides some theorems and algorithms for certification. Experiments results on synthetic datasets (Cifar10 and imagenette) and a real-world medical dataset are also reported.

**Strengths:**

1. The paper studied an interesting problem, where misclassification to different classes will trigger different costs.
2. The paper claims that this is the first work to apply randomized smoothing to cost-sensitive learning setting.
3. The paper provides some theorems for deriving the robustness radius (thm 3.2) and shows the situations where cost-sensitive robust radius is strictly bigger than regular robust radius (thm 3.3).
4. Experiments showed some improvements over the baseline methods.

**Weaknesses:**

1. Although providing theorem 3.2 for computing robust radius, the proof of this theorem is a straightforward generalization of Theorem 1 in [Cohen et al 2019], hence raising a novelty concern. Note that this limited novelty doesn't affect the reviewer's rating on this work.
2. This method is designed to handle the cost-sensitive setting, for example, medical classification as the paper claimed. However, the improvement in medical setting is not as noticeable as in synthetic setting (Cifar10 and imagenette).

**Questions:**

As mentioned in weakness 2, can the author perhaps explain why the proposed method has less improvement on HAM10k compared with Cifar10 and imagenette?

---

### Official Review · Reviewer_B2mS · 2023-11-01

**Soundness:** 2 fair
**Presentation:** 2 fair
**Contribution:** 2 fair
**Rating:** 3
**Confidence:** 4

**Summary:**

The paper delves into adversarially robust classifiers in cost-sensitive contexts, where adversarial transformations are assigned varying importance through a binary cost matrix. Many existing solutions either can't guarantee robustness or aren't scalable. This research leverages randomized smoothing, a method known for its scalability, to certify robustness in these cost-sensitive scenarios. The authors introduce a new metric, the cost-sensitive certified radius, and design an algorithm to train classifiers with this robustness in mind. Their approach is validated with experiments on both image datasets and a real-world medical dataset, achieving enhanced robustness without sacrificing accuracy.

**Strengths:**

1. The paper is well-written with a clear motivation.

2. The topic is important.

**Weaknesses:**

1. The paper appears to be a straightforward adaptation of Cohen et al. [1] to a cost-sensitive context, limiting its novelty. Algorithm 1 seems to selectively ignore classes based on the cost matrix, and the training approach in Section 4.2 appears to be a variation of MACER. If my assessment is incorrect, I'd appreciate clarification in the rebuttal about the unique challenges faced when integrating randomized smoothing [1] and MACER into the cost-sensitive framework.

2. The paper lacks comparisons with several key baselines [2,3,4]. It's essential to address these well-established methods, especially since SmoothMix reportedly outperforms MACER in [4].

[1] Cohen et al., Certified Adversarial Robustness via Randomized Smoothing, ICML 2019\
[2] Salman et al., Provably Robust Deep Learning via Adversarially Trained Smoothed Classifiers, NeurIPS 2019\
[3] Jeong et al., Consistency Regularization for Certified Robustness of Smoothed Classifiers, NeurIPS 2020\
[4] Jeong et al., SmoothMix: Training Confidence-calibrated Smoothed Classifiers for Certified Robustness, NeurIPS 2021

**Questions:**

1. Refer to Weakness 1.

2. Given the plethora of papers focused on refining the base classifier's training, why was MACER chosen as the foundation? Does MACER offer specific advantages for cost-sensitive training? If so, could you elucidate what aspects of MACER enhance its applicability to the cost-sensitive context?

3. I observed that your method's distinction from MACER lies in its consideration of misclassified samples, indicated by $R_{c-s}<0$. You assert on Page 17 about achieving a superior balance between accuracy and cost-sensitive robustness. Can your enhanced MACER also boost the original certified robustness presented in [1]? Specifically, could you provide $Rob_{std}$ in Table 5? Such improvements might stem from optimizing misclassified samples and may not be inherently tied to cost-sensitive attributes.